# Biochemical and Behavioral Characterization of IN14, a New Inhibitor of HDACs with Antidepressant-Like Properties

**DOI:** 10.3390/biom10020299

**Published:** 2020-02-14

**Authors:** Heidy Martínez-Pacheco, Ofir Picazo, Adolfo López-Torres, Jean-Pascal Morin, Karla Viridiana Castro-Cerritos, Rossana Citlali Zepeda, Gabriel Roldán-Roldán

**Affiliations:** 1Laboratorio de Neurobiología Conductual, Departamento de Fisiología, Facultad de Medicina, Universidad Nacional Autónoma de México, Ciudad de México 04510, Mexico; heidymartinezp@gmail.com (H.M.-P.); jpmorin@comunidad.unam.mx (J.-P.M.); 2Sección de Estudios de Posgrado e Investigación de la Escuela Superior de Medicina del Instituto Politécnico Nacional, Plan de San Luis y Salvador Díaz Mirón s/n, Casco de Santo Tomás, Ciudad de México 11340, Mexico; rifo99mx@gmail.com; 3Instituto de Química Aplicada, Universidad del Papaloapan, Campus Tuxtepec, Circuito Central 200, Parque Industrial, Tuxtepec, Oaxaca 68301, Mexicokarlaviri8@hotmail.com (K.V.C.-C.); 4Centro de Investigaciones Biomédicas, Universidad Veracruzana. Av. Dr. Luis Castelazo Ayala s/n. Col., Industrial Ánimas, Xalapa, Veracruz 91190, Mexico; rzepeda@uv.mx

**Keywords:** chromatin remodeling, antidepressants, toxicity, forced-swimming test, drug design, epigenetics, elevated T maze, class I HDAC

## Abstract

Evidence suggests that histone deacetylases (HDACs) inhibitors could be used as an effective treatment for some psychiatric and neurological conditions such as depression, anxiety and age-related cognitive decline. However, non-specific HDAC inhibiting compounds have a clear disadvantage regarding their efficacy and safety, thus the need to develop more selective ones. The present study evaluated the toxicity, the capacity to inhibit HDAC activity and antidepressant-like activity of three recently described class I HDAC inhibitors IN01, IN04 and IN14, using *A. salina* toxicity test, *in vitro* fluorometric HDAC activity assay and forced-swimming test, respectively. Our data show that IN14 possesses a better profile than the other two. Therefore, the pro-cognitive and antidepressant effects of IN14 were evaluated. In the forced-swimming test model of depression, intraperitoneal administration of IN14 (100 mg/Kg/day) for five days decreased immobility, a putative marker of behavioral despair, significantly more than tricyclic antidepressant desipramine, while also increasing climbing behavior, a putative marker of motivational behavior. On the other hand, IN14 left the retention latency in the elevated T-maze unaltered. These results suggest that novel HDAC class I inhibitor IN14 may represent a promising new antidepressant with low toxicity and encourages further studies on this compound.

## 1. Introduction

Epigenetic mechanisms involving chromatin-modifying enzymes have been implicated in the pathophysiology of mental and neurodegenerative diseases such as depression, anxiety and cognitive deficits [1,2,3,4,5], as well as in the therapeutic mechanisms of some antidepressants [6,7,8,9,10,11,12]. Amongst epigenetic mechanisms, histone acetylation—the addition of an acetyl group in the N-terminal lysine residues in the nucleosomal core of histone proteins [13,14]—has been highly implicated in neuroplasticity [15]. Histone acetylation is associated with the opening of chromatin structure that facilitates the binding of transcription activating protein complexes that modulate gene expression [16], while deacetylation leads to a transcriptionally inactive chromatin state [17,18]. 

Histone deacetylase (HDAC) enzymes include subtypes comprising class I (HDAC1, 2, 3 and 8) and class II (HDAC 4–7, 9 and 10). Recent findings indicate that the activity of specific class I HDAC enzymes may be altered in psychiatric and neurodegenerative disorders and may therefore constitute a target for novel pharmacological treatments. The HDAC inhibitors targeting class I, sodium butyrate and MS-275 were reported to produce antidepressant-like effects [19,20,21]. MS-275 was also associated with increased histone acetylation and decreased levels of HDAC2 [19,20]. Recent studies by two distinct groups have demonstrated that neuron-specific over-expression of HDAC2 impaired memory formation in adult mice, whereas HDAC2 deficiency resulted memory facilitation, similar to that induced by the treatment with nonselective HDAC inhibitors, suggesting that HDAC2 inhibition may be a useful strategy in the treatment of some cognitive impairments associated with psychiatric diseases [22,23].

As efficacy and safety data from non-selective HDAC inhibitors emerge, the inquiry remains as to whether it will be advantageous to develop class I-selective inhibitors to improve the therapeutic index over non-selective inhibitors. In the present study, three recently designed HDAC inhibitors [24] were characterized in animal models. First, their toxicity (LD_50_) was examined by means of *A. salina* test, which is one of the most widely approved test organisms available for toxicity testing [25]. A specific in vitro ELISA-based test was then used to explore the HDAC selectivity of these three HDAC inhibitors. Based on the results obtained in these assays, we selected IN14 and evaluated its effects on behavior. Its antidepressant-like properties were evaluated in the forced-swimming test (FST), a putative model of depression, while the elevated T-maze (ETM) was used to explore its actions on learning and memory. 

## 2. Materials and Methods 

### 2.1. Animals 

Adult young male CD1 mice weighing 22 to 25 g (*n* = 74) were obtained from the colony of the Facultad de Medicina, Universidad Nacional Autónoma de México (UNAM). The animals were simple randomized to the treatment groups using the random number generator (Rand function) of MATLAB software and housed in a temperature-controlled room (22 ± 1 °C) with a 12 h light-dark cycle (lights on at 07:00 h) and ad libitum access to food and water. Experiments were performed in accordance with the protocols approved by the Committee on the Use of Live Animals in Teaching and Research of the UNAM (FM/DI/036/2017), which comply with the “International Guiding Principles for Biomedical Research Involving Animals,” Council for International Organizations of Medical Sciences, 2010. Efforts were taken to minimize animal suffering throughout the experiments. Moreover, all the experiments were performed in a double-blind manner.

### 2.2. Chemicals and Reagents

As mentioned, the compounds IN01, IN04 and IN14 were previously synthesized and characterized by our group (Figure 1A) [24]. The compounds were characterized and the yield and purity were determined using thin layer chromatography and spectroscopic techniques (^1^H and ^13^C quantitative Nuclear Magnetic Resonance (qNMR) and Electrospray Ionization (ESI) high resolution mass spectrometry) (Figure 1B). Sodium phenylbutyrate (PB), Desipramine hydrochloride (DMI), Pentobarbital, potassium dichromate (K_2_Cr_2_O_7_) and MS-grade ammonium formate were purchased from Sigma, St Louis, MO, USA. MS-grade methanol and formic acid were purchased from Merck, S. A de C.V. (Naucalpan de Juárez, México). Deionized water (resistivity 18.2 MΩ-cm) for sample pre-processing and mobile phase preparation was obtained from a water purification system (ThermoFisher Scientific; Naucalpan de Juárez, México).

### 2.3. Artemia Salina Toxicity Test

#### 2.3.1. Hatching of Artemia Salina

*A. salina* cysts (Eclosion azul®) were obtained at a local aquarium and hatched in seawater (3%). Artificial seawater was prepared by dissolving salt for the aquarium (San-Halita, Biomaa; Jilotzingo, México) in deionized water and stirred for 24 h under aeration and then filtered through 30 μm Millipore cellulose filters before use. Approximately 0.1 g of cleansed *Artemia* cysts were incubated in 1 L of seawater (pH 8.5–9) at 25 ± 2 °C with a light intensity of 8.6 Klux. Air was pumped through the bottom of the container to prevent settling of cysts. Hatching was completed within 15 to 24 h, however, only the nauplii, which hatched from the cysts during the 24 h of incubation, were used to start the toxicity tests.

#### 2.3.2. Toxicity of HDAC Inhibitors to *A. Salina*

*A. salina* nauplii were exposed to 0.1, 1, 10, 100, 300 and 9000 ppm solutions of PB, IN01, IN04 and IN14. For all HDAC inhibitors, the procedure of toxicity tests was identical. Crustaceans were chemically exposed in a 48 h toxicity test, following the guideline for *Artemia* toxicity screening test (Artoxkit, ECOtest, Spain). Three replicates were prepared per test concentration. We added 10 nauplii per well in the well plates and incubated in the dark at 25 °C for 48 h. The numbers of surviving nauplii in each well were counted under a stereoscopic microscope (SZ-PT, Olympus) after 48 h. The experiments were conducted in triplicate for each concentration. To compare the sensitivity of the animals used in the different tests, a toxicity test with the reference compound potassium dichromate (K*_2_*Cr*_2_*O*_7_*) prepared in seawater was performed with each set of toxicity tests. The % mortality vs. log [K*_2_*Cr*_2_*O*_7_*] (ppm) was plotted to obtain the classic sigmoidal dose-response graph, where the inflection point represents the LC_50_.

### 2.4. In vitro ELISA-Based HDAC Activity Assay

HDAC activity assay was determined using the fluorometric *in vitro* HDAC activity assay FLUOR DE LYS® (Enzo Life Sciences, Inc.) according to manufacturer’s protocol, and background subtracted relative fluorescence unit (RFU) counts were acquired by fluorometer Molecular Devices, GeminiXS (360 nm excitation, 450 nm emission). In this experiment the inhibitors were used as follows: 5 nM IN01, 5 nM IN04, 5 nM IN14, 0.5 nM TSA and 5 nM PB. To minimize variability the inhibition assays were performed as technical triplicate.

### 2.5. Pharmacokinetics of IN14

Considering the toxicity data (compound with the lowest toxicity) and HDAC activity (compound that most inhibited HDAC activity), we decided to determine if compound IN14 crossed the blood-brain barrier (BBB) using the following methodology.

#### 2.5.1. Mice Treatment

Mice (*n* = 5) received either a unique injection or a daily injection for five consecutive days of saline solution or IN14 (100 mg/kg; i.p.). On behavioral test day, mice were administered 1 h prior the assay. 

#### 2.5.2. Rodent Biofluid Harvesting and Purification

An independent group of animals (*n* = 5) was used to determine the levels of IN14 within plasma, urine and cerebrospinal fluid 24h after last administration of the drug. Control or treated mice were euthanized with deep anesthesia using an injection of Pentobarbital (50 mg/kg, i.p.). The skin at the back of the neck, the chest and the lower abdomen was shaved and alcohol pads were used to clean the hairless skin. A horizontal incision opened the lower abdominal wall, and a 1 mL syringe was used to withdraw urine directly from the bladder. A longitudinal incision in the back of the neck skin was then made through which the foramen magnum was uncovered. A 1 mL syringe was used to puncture through the dura and withdraw cerebrospinal fluid (CSF). Finally, through an incision in the chest, blood was withdrawn from the right atrium using an ethylene-diamine-tetra-acetic acid (EDTA) 5 mL syringe. The blood samples were centrifuged at 1000× *g* for 1 min at 4 °C and plasma was collected. One volume of mobile phase (10 mM HCOONH_4_, pH = 4.1) and two volumes of MeOH were added to the plasma, urine and CSF samples, which were vortex mixed for 1 min and incubated for 2 h at −20 °C. Subsequently, samples were centrifuged at 15,000 g for 10 min at 4 °C and the supernatants were collected. Samples were then stored at 20 °C until use.

Extracts were cleansed with a Supel^TM^-Select HLB SPE Tube bed wt., vol. 1 mL 30 mg; (Supelco). Before loading the plasma, urine and cerebrospinal fluid samples onto the SPE system, the cartridge was activated with 1.5 mL MeOH and conditioned with 1.5 mL MeOH 50% (*v*/*v*). Samples were loaded onto the SPE tube, eluted with 1.5 mL of 50 % MeOH, totally recovered and evaporated to dryness. Samples were resuspended in 150 μL of mobile phase A and kept at –20 °C until chromatographic analysis.

#### 2.5.3. UPLC–ESI-TOF-MS Analysis

The compound IN14 was monitored in biological samples using a chromatographic system ultra-performance liquid chromatography (UPLC) Acquity I-Class (Waters) coupled to Q-TOF mass spectrometer system Synapt G2-Si (Waters) equipped with ESI source. One μL of extract was injected onto a column Luna omega C18 1.6 µm (2.1 mm × 150 mm, Phenomenex) heated at 40 °C, using the mobile phase A: 10 mM HCOONH*_4_* pH 4.1, and mobile phase B: MeOH at a flow rate of 0.1 mL/min with the following gradient elution method: 0–3 min, 5% B; 3–5 min, 20 % B, 5–8 min, 20 % B, 8–8.1 min, 5 % B, 8.1–10 min, 5 % B.

Detection was carried out in positive mode using the following settings: capillary 3 kV, temperature 120 °C, sampling cone 40, source offset 70, cone gas flow 50 L/h, desolvation gas flow 500 L/h, desolvation temperature 350 °C. Full-scan mass spectra were acquired in centroid mode from 100–1200 m/z. The mass spectrometer was calibrated using NaI and leucine enkephalin was used as the lock mass. MassLynx software version 4.1 (Waters) was used to control the instrument and data analysis.

### 2.6. Behavioral Assays

All behavioral experiments were carried out between 10:00 and 14:00 h, with a time interval of 1 h between the last dose (vehicle or IN14) and the behavioral test. The sessions were recorded using a video camera. Every test day, mice were habituated for 1 h prior to testing by placing their cage in the experimental room with no water bottle or feeder bin for 5 min.

#### 2.6.1. Treatment

Mice were administered with: vehicle (saline solution, i.p.), IN14 (100 mg/kg, i.p.), PB (100 mg/kg, i.p.) or DMI (30 mg/kg, i.p.) for 5 days 1h before behavioral tests. Day 5 of administration was considered the first day of the test. The dose of IN14 was chosen based on preliminary behavioral experiments showing that 100 mg/kg/day was as effective as higher doses tested (200 and 400 mg/kg) [26].

#### 2.6.2. Forced-Swimming Test

The Forced-Swimming Test (FST) test employed in this study was based on the originally published procedure [27] for rats. We made some modifications to test mice according to Farzin and Mansouri [28]. Briefly, mice were placed individually in a cylindrical glass tank (25 cm height × 13 cm diameter) filled with water (25 ± 1 °C) to a depth of 10 cm for 15 min. Immediately after the 15-min swim, mice were removed from the tank, dried with a towel and put in a warming cage (37 °C) that contained a heating pad covered with towels for 15 min. Mice were then returned to their home cage. The forced swim occurred between 10:00 and 14:00 h. On the following day (24 h after the first swimming session), mice were placed in the same water tank for 5 min. The entire process was videotaped using a digital camera and behavioral activity was recorded during a period of 5 min per session. Each session was divided in 60 accounts of 5 s during which three behavioral responses were recorded: immobility, motions necessary to hold their head above water; swimming, crosses from one quadrant to another; and climbing, actions to escape by climbing the cylinder walls. The duration of immobility was regarded as an indication of helplessness and was used as an index of depression-like behavior as reported previously [29].

#### 2.6.3. Elevated T Maze 

The Elevated T maze (ETM) apparatus was made of acrylic, had three arms of equal dimensions (33 cm × 5 cm) and was elevated 50 cm above the floor. One arm, enclosed by walls 25 cm high, was perpendicular to the two opposed open arms. Behavioral testing was carried out in a soundproof room with an illumination level maintained at 100 lux. Independent groups of control and treated animals (*n* = 8) were trained. During the acquisition session each mouse was placed at the end of the enclosed arm of the maze and the time to exit this arm and enter with all four paws to any of the open arms was recorded (acquisition latency, AL). The same maneuver was then repeated in the subsequent trials at intervals of 1 min (acquisition latencies AL1, AL2, AL3...) until the whole group reached the learning criterion, i.e., an AL of 180 s. In addition, a cutoff time of 180 s was established for the acquisition latencies; thus if a mouse did not leave the enclosed arm in this period, the trial was ended and a latency of 180 was assigned. To be on an open arm is an aversive experience since rodents have an innate fear to height and openness [30,31]. Thus, when the animal is repeatedly placed inside the enclosed arm and allowed to explore the maze they acquire an inhibitory avoidance to the open arms. In order to verify that the animals did not have any motivational or motor impediment that prevented them from performing the task, once each mouse had reached the learning criterion it was placed at the end of one open arm on which they had been pre-exposed and the latency to leave this arm and enter to the enclosed arm with all four paws was recorded in a single trial (escape latency, EL). Forty-eight hours later long-term memory was evaluated in a single-trial identical to those of the acquisition session where the time to leave the enclosed arm was recorded (retention latency, RL).

### 2.7. Data Analysis

The Anderson-Darling Normality test was performed in all groups. All data are expressed as the mean ± standard error of mean (SEM). Non-linear regression curve fit toll was used to obtain the LC_50_ of all compounds tested. The FST, which did not pass normality test, was analyzed with Kruskal-Wallis tests followed by Dunn’s multiple comparisons test to compare each group with the vehicle. All other experiments were analyzed using ANOVAs with post-hoc Tukey tests where appropriate. The GraphPad Prism® software version 6.01 was used to perform all analyses. Differences between groups were considered statistically significant if *p* < 0.05.

## 3. Results

### 3.1. LC_50_ of IN01, IN04 and IN14

The new compounds and the HDAC inhibitor PB showed a concentration-dependent increase in the toxicity for the 24-h-old nauplii. IN01 showed significantly higher toxicity compared to IN04, IN14 and PB (*p* < 0.001). The concentration of 900 mg/L of IN01, IN04, IN14 and PB was lethal to nauplii. The LC_50_ value of IN01, IN04, IN14 and PB was found to be 166.0 mg/L, 393.9 mg/L, 327.0 mg/L and 295.5 mg/L, respectively with 95% confidence. No sign of mortality was observed in the control (seawater-only) group. The concentration–response curves obtained for each compound tested on *A. salina* are plotted in Figure 2.

### 3.2. Inhibitory Activity of IN01, IN04 and IN14 Against Class I HDACs.

To evaluate the degree of HDAC activity inhibition by each of the compounds, we next measured HDAC activity with FLUOR DE LYS®. A one-way ANOVA showed significant differences between groups (F5, 12 = 880.7; *p* < 0.001). The post-hoc Tukey test revealed that IN14 significantly inhibited HDAC activity (*p* < 0.001; Figure 3); interestingly, the compound IN01 increased HDAC activity (*p* < 0.05; Figure 3). Given that the most potent class I HDAC inhibition was obtained with IN14, we decided to use this compound for the next experiments.

### 3.3. Compound IN14 Crosses the BBB 

Presence of IN14 in different biofluids from five mice was monitored by liquid chromatography–mass spectrometry (LC-MS). Figure 4 shows representative extracted ion chromatograms (EIC) of molecular ion *m*/*z* 236—the protonated form of IN14—(Figure 4A) from IN14, samples of plasma, urine and cerebrospinal fluid of mice treated for one or five days. IN14 was detected in plasma, urine and cerebrospinal fluid since the first day of administration, suggesting its easy distribution.

### 3.4. Behavioral Assays

We next analyzed the effects of IN14, PB, and DMI on distinct parameters of the FST test. Kruskal-Wallis tests showed that swimming (H = 26.59, *p* < 0.001), climbing (H = 26.63, *p* < 0.001) and immobility (H = 25.23, *p* < 0.001) behaviors varied significantly between groups (*n* = 8) (Figure 5). When analyzing specific group differences in active behaviors, Dunn’s tests revealed that swimming did not differ between IN14 and vehicle groups (*p* = 0.9062), while mice treated with PB had a tendency to swim more, although the difference did not reach significance (*p* = 0.062). Interestingly, swimming was significantly decreased in the DMI group compared to vehicle (*p* < 0.05). Meanwhile, treatment with either IN14 or DMI increased climbing behavior compared to vehicle (*p* < 0.001 in both cases), whereas no significant difference was observed between PB and vehicle groups (*p* > 0.25). Finally, Dunn’s tests revealed that both IN14 and PB significantly decreased the immobility time, a putative indicator of helplessness, in comparison to vehicle (*p* < 0.001 in both cases) while treatment with DMI did not produce any significant effect (*p* > 0.19).

In the ETM test, two-way repeated measures ANOVA yielded no effect of treatment (F2, 24 = 0.8338, *p* = 0.4466) nor treatment x trial interaction (F14, 168 = 1.122, *p* = 0.3419) but did unveil a significant main effect of trial (F7, 168 = 177.6, *p* < 0.001; Figure 6A), indicating that all groups (*n* = 9) performed similarly in this task. Figure 6A shows that during the acquisition, mice of all three groups reached the criterion of learning by remaining 180 s in the closed arm after seven training trials. Long-term memory test 48 h after training revealed a near perfect performance in all groups. Furthermore, there were no significant differences in escape latencies (Figure 6B), ruling out the possibility that motor impediment could have compromised mice performance during the test (one-way ANOVA, F2, 24 = 0.662, *p* = 0.525).

## 4. Discussion

The present study evaluated the biochemical properties and behavioral effects of recently designed [24] HDAC inhibitors. We first performed a lethality bioassay and found that among the compounds tested, IN04 and IN14 appeared to be the least toxic. For a substance to be considered highly toxic the LC_50_ has to be reached at concentrations of 1–10 μg/mL [32]. All three compounds tested herein had LC_50_, which shows that they are safe under the parameters of the *A. salina* test.

The capacity of IN14 (AFU: 16993) to inhibit class I HDACs was comparable to that of TSA (AFU: 18602), a potent HDAC inhibitor, which induces cell proliferation, cell differentiation and apoptosis *in vitro* [33]. IN04 (AFU: 31994), on the other hand, has inhibition effectiveness similar to that of PB (AFU: 33402), a reversible inhibitor of class I and II HDACs [34]. In contrast, IN01 potentiated the activity of the HDAC enzymes, which suggests that this compound could be useful in models where HDAC hyperactivity is required.

To evaluate possible effects of IN14 on behavior, we chose CD1 mice because of their higher genetic variability as an outbred mouse strain, which is probably desirable when testing new compounds with possible translational applications. Furthermore, CD1 mice have been used in inducible models of depression [35,36,37] and are responsive to antidepressants such as imipramine [35] and desipramine [38]. Our results also show that intraperitoneal injection of IN14 decreased forced-swimming-induced behavioral despair, as observed by a decrease in immobility in the FST task within five days of treatment, similar to what is observed with PB. Strikingly, both PB and IN14 were superior to tricyclic antidepressant DMI at reducing immobility. These findings are at odds with previous observations on hybrid 129SvEv/C57Bl/6 mice that showed no effect of chronic treatment with class I inhibitor PB on the FST at a dose that did increase histone acetylation [39]. However, in a chronic social defeat mouse model of depression, it was reported that the administration of the HDAC inhibitors suberoylanilide hydroxamic acid (SAHA) and MS-275 in the nucleus accumbens resulted in increased acetylation of the histone H3K14; an effect accompanied by a significant antidepressant-like effect, including decreased immobility in the FST [19]. Furthermore, SAHA was shown to decrease immobility in the FST in Crtc1 knockout mice, who exhibit a depressive-like phenotype, as well as in C57BL/6 wildtypes [40]. These discrepancies are likely due to differences in the characteristics of the specific class I HDAC inhibitors employed as well as variations in the behavioral protocols.

Also noteworthy is the observation that IN14 was more efficient than PB at increasing climbing behavior whereas it left swimming behavior unaltered, suggesting a more potent and specific effect in promoting motivational behavior and, at the neurochemical level, this may point to a long-term enhancement of noradrenergic rather than serotoninergic transmission [41]. Furthermore, multiple studies have reported that the inhibition of HDACs increases motivation, the search for reinforcements and reduces symptoms in models of depression when administered in the hippocampus, prefrontal cortex, amygdala or the nucleus accumbens [42]. However, these actions are influenced by differences in HDAC inhibitor potency, specificity, tissue distribution and effectiveness at crossing the BBB [43]. 

The overexpression of HDAC enzymes in the forebrain has been related to cognitive deficits. For instance, HDAC2 was reported to negatively regulate the structural and functional synaptic plasticity, as well as memory formation in the hippocampus [22]. In line with this, the majority of studies searching the histone acetylation-memory relationship have used HDAC inhibitors to demonstrate that the increase in lysine acetylation resulting from the inhibition of HDACs enhances cognition in animal models of neurodegenerative diseases [1,2,3,4,5,44]. The current study did not find any alteration or beneficial effect of IN14 and PB on learning as assessed by the ETM task in contrast with an earlier study reporting that HDAC2 knockout mice performed significantly better in this task [22]. This discrepancy could be due to the fact that IN14 inhibit all class I HDACs, not just HDAC2 and some HDAC inhibiting compounds such as TSA have been reported to have an anxiolytic effect [45,46]. Given the anxiogenic nature of the ETM task [47], it is possible that an anxiolytic effect of IN14 and/or PB could have masked a memory enhancing effect. In addition, in the current study only intact, cognitively unimpaired mice were used; therefore, it remains to be demonstrated whether IN14 may facilitate learning and memory in cognitively impaired animals, a possibility that we are currently assessing. Finally, class I HDAC inhibitor sodium butyrate was recently demonstrated to promote consolidation and reconsolidation of spatial memory in intact mice when administered after a weak training that normally allowed for short-term but not long-term memory retention [48]. Further studies are needed to determine whether IN14 may improve learning and memory and if so, under what circumstances and which step of the memory formation process.

## 5. Conclusions

Taken together, the data presented herein suggest that the new agent IN14, which inhibits all class I HDACs *in vitro*, may represent a new anti-depressant drug candidate since it has low toxicity, is permeable to the BBB and shows clear actions within five days of treatment initiation. Further in vivo studies are needed to evaluate the therapeutic potential of this promising new compound. 

## Figures and Tables

**Figure 1 biomolecules-10-00299-f001:**
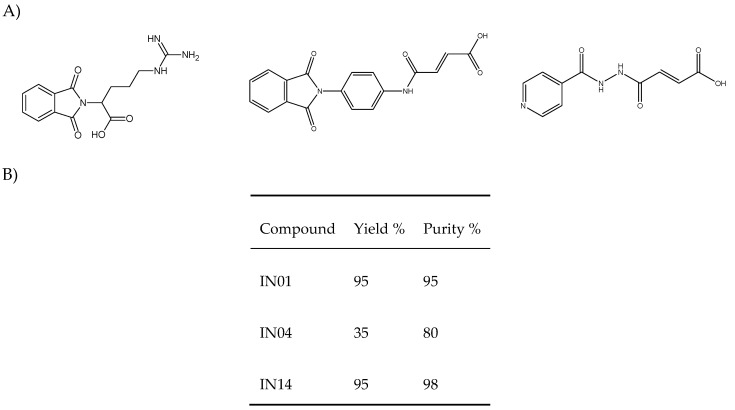
(**A**) Chemical structures of compounds IN01 (left), IN04 (center) and IN14 (right). (**B**) Yield and purity of the new histone deacetylases (HDAC) inhibitors synthetized.

**Figure 2 biomolecules-10-00299-f002:**
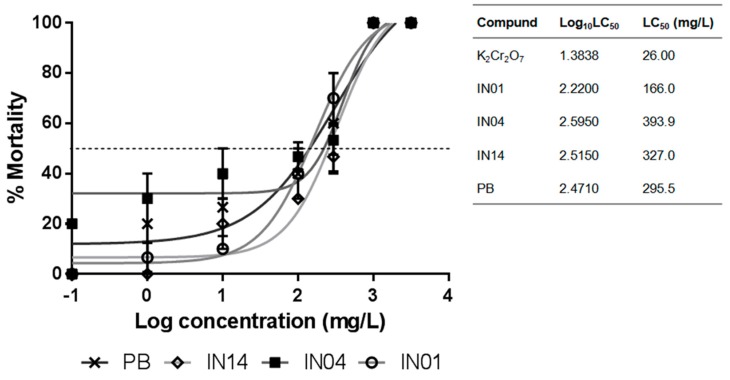
Percentage of mortality of *A. salina* exposed to HDAC inhibitors. Acute toxicity (LC_50_) of IN01, IN04, IN14 and sodium phenylbutyrate (PB) on *A. salina* in seawater medium. The concentrations of the compounds (IN01, IN04 and IN14) and the HDAC inhibitor, phenylbutyrate (PB), are expressed in log10 concentration. Each point represents the mean ± SEM of 30 nauplii.

**Figure 3 biomolecules-10-00299-f003:**
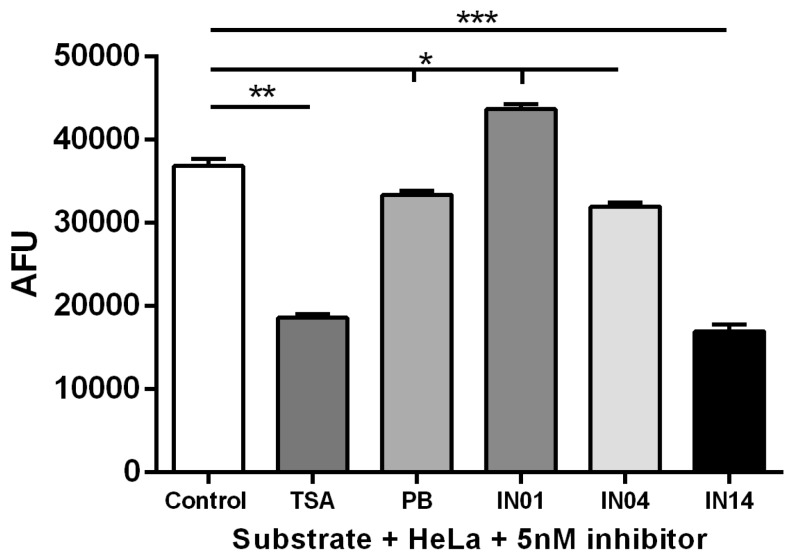
IN14 inhibits HDAC activity in HeLa Nuclear Extract. Data are expressed as mean ± SEM. Experiments were performed by technical triplicate. HDAC activity (arbitrary fluorescent units [AFU]); PB: phenylbutyrate; TSA: Trichostatin A (positive control); IN01-IN04-IN14: synthesized compounds. * *p* < 0.05, ** *p* < 0.01, *** *p* < 0.001 vs. control.

**Figure 4 biomolecules-10-00299-f004:**
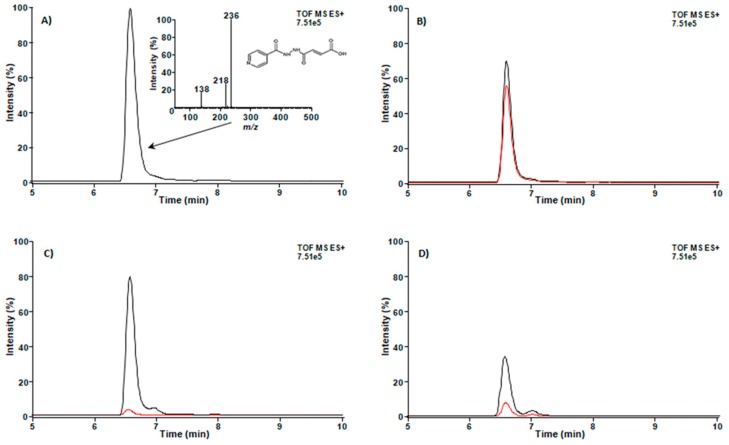
Detection of IN14 by LC-MS. (**A**) IN14 chromatogram and electrospray ionization (ESI) (+) mass spectrum. Extracted ion chromatograms (EIC) of *m*/*z* = 236 in samples of cerebrospinal fluid (**B**), plasma (**C**), and urine (**D**). Mice (*n* = 5) were treated with IN14 for one day (red line) and five days (black line), fluid extraction and LC-MS analysis are described in the Materials and Methods section.

**Figure 5 biomolecules-10-00299-f005:**
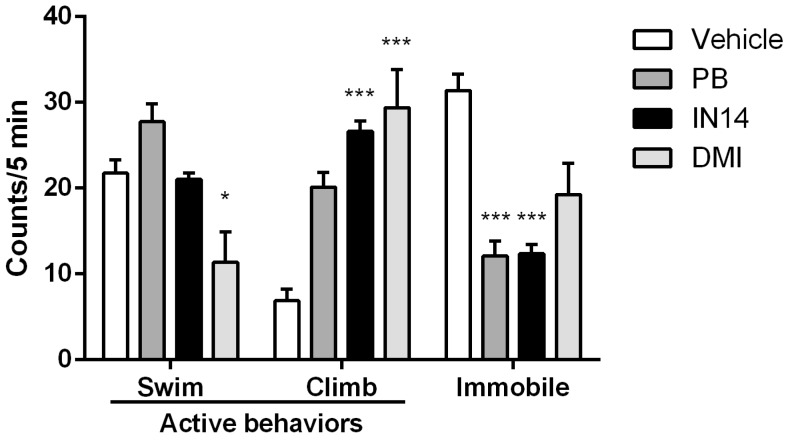
Performance of mice treated with vehicle, IN14, phenylbutyrate (PB) or Desipramine hydrochloride (DMI) during forced swimming test. Bars indicate the incidence of swimming (swim), climbing (climb) and immobility (immobile) during a 5-min test. Data are expressed as mean ± SEM (*n* = 8). * *p*  <  0.05; *** *p*  <  0.001 vs. vehicle.

**Figure 6 biomolecules-10-00299-f006:**
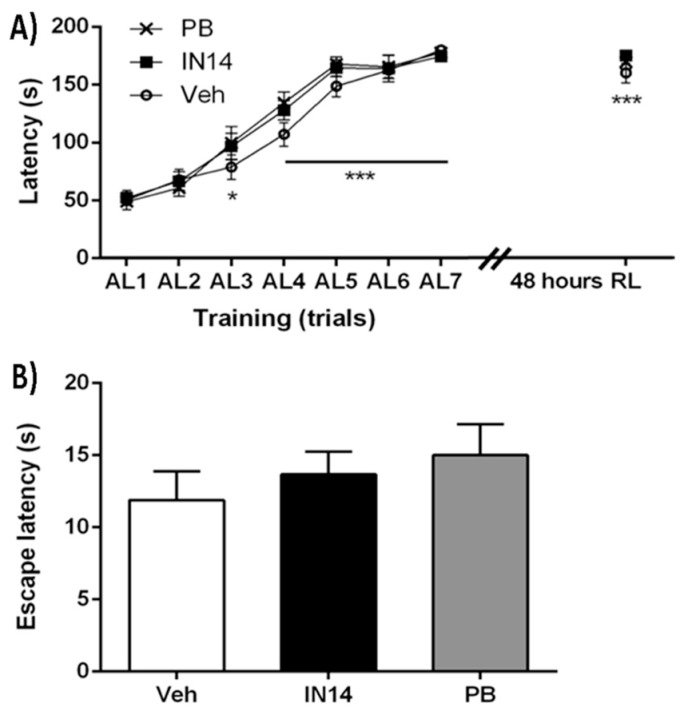
Pre-training intraperitoneal administration of phenylbutyrate (PB) or IN14 does not affect memory performance. (**A**) Elevated T-maze learning curves of intact mice during the training session and retention latencies. * *p* < 0.05, *** *p* < 0.001 vs. the first acquisition latency (AL1), *n* = 9. (**B**) Escape latency from the open arm in the elevated T-maze (ETM) test. No significant differences were observed among different groups. ns: *p* = 0.5250, EL: escape latency; RL: retention latencies.

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
