# Peer review of "Biochemical and Behavioral Characterization of IN14, a New Inhibitor of HDACs with Antidepressant-Like Properties"

_biomolecules, 2020, doi:10.3390/biom10020299_

Round 1

Reviewer 1 Report

In "Biochemical and behavioral characterization of IN14, a new inhibitor of HDACs with antidepressant-like properties", the authors characterize the toxicity, HDAC inhibition efficacy, BBB permeability of IN14, a newly synthesized HDAC inhibitor. The authors also demonstrate that IN14 produces behavioral responses that mimic/surpass those of antidepressant desipramine in the forced swim test without altering cognitive functions. This manuscript would be of interest to the readership of Biomolecules and could be a good addition to the literature. I have several concerns that need to be addressed in the revised version.

The authors should specify the time point at which the biofluids were collected and behavioral tests were carried out (i.e. the interval between drug treatment and collection/test) in the Materials and Methods section.   The authors should clarify the protocol (e.g. drug concentration, treatment timeline, etc.) for the HDAC inhibition test in the Materials and Methods section. Have the authors tried other concentrations in the HDAC inhibition experiment?  The authors should clarify why 100 mg/kg IN14 dosage was chosen in the behavioral experiments. It looks like there are comparable IN14 levels in cerebrospinal fluid in acute vs. 5 days treatment group. Have the authors performed behavioral tests after acute IN14 treatment? 

Author Response

Reviewer 1

In "Biochemical and behavioral characterization of IN14, a new inhibitor of HDACs with antidepressant-like properties", the authors characterize the toxicity, HDAC inhibition efficacy, BBB permeability of IN14, a newly synthesized HDAC inhibitor. The authors also demonstrate that IN14 produces behavioral responses that mimic/surpass those of antidepressant desipramine in the forced swim test without altering cognitive functions. This manuscript would be of interest to the readership of Biomolecules and could be a good addition to the literature. I have several concerns that need to be addressed in the revised version.

We really appreciate the observations and suggestions made; we believe were very useful to improve the quality of our work.

The authors should specify the time point at which the biofluids were collected and behavioral tests were carried out (i.e. the interval between drug treatment and collection/test) in the Materials and Methods section.

We agree with the reviewer that these points need clarification. We therefore performed the following changes it in the Materials and Methods section:

In section 2.5.2. Mice treatment, lines 146-7 we added: On behavioral test day, mice were administered 1h prior the assay. “

The behavioral tests were carried out between 10 and 14h, with a time interval of 1 h between the last dose (vehicle or IN14) and the behavioral test, as mentioned in lines 183-4.

Finally, the collection of fluids was performed 24h after the last administration of IN14, after sacrifice by pentobarbital overdose, in order to detect the presence and levels of the drug in urine, plasma and CSF. Therefore, those animals were not evaluated in the behavioral tests. In lines 149-51 we added the sentence: “An independent group of animals (n=5) was used to determine the levels of IN14 within plasma, urine and cerebrospinal fluid 24h after last administration of the drug.

The authors should clarify the protocol (e.g. drug concentration, treatment timeline, etc.) for the HDAC inhibition test in the Materials and Methods section.

As suggested by the reviewer, we clarified that we performed the inhibition test following the instructions of the manufacturer, provided with the assay kit; we also included the concentration of all drugs tested in the new version. Please refer to the 2.6.1 “Treatment” section.

Have the authors tried other concentrations in the HDAC inhibition experiment?

We have not. However, since TSA inhibits Class I HDAC activity in nM range (Furumai et al 2001. PNAS. 98(1):87–92. doi:10.1073/pnas.011405598), we tested our compounds in the same nM range (100 nM), obtaining an in vitro inhibition activity similar to TSA (figure 3).

The authors should clarify why 100 mg/kg IN14 dosage was chosen in the behavioral experiments.

Certainly, the reviewer is right. In a preliminary behavioral study, we tested the same five-days treatment reported here using IN14 100, 200 and 400 mg/kg/day; the results of these experiments showed that all three doses had similar effects on spontaneous locomotor activity and the FST. Therefore, we decided to use the lower effective dose in this work. We now include a sentence in the manuscript to clarify this point. Please refer to 2.6.1 “Treatment” section.

It looks like there are comparable IN14 levels in cerebrospinal fluid in acute vs. 5 days treatment group. Have the authors performed behavioral tests after acute IN14 treatment?  

The reviewer's observation is correct. We have noticed that, but we did not evaluate an acute single dose of IN14 treatment in the present study. However, in a separate set of experiments aimed at comparing different HDAC inhibitors on memory impairment induced by acute stress, we have tested a two 100 mg/kg/day IN14 administration obtaining similar effective results than a five-days treatment. The manuscript containing these results is currently under preparation.

Reviewer 2 Report

The manuscript “Biochemical and behavioral characterization of IN14, a new inhibitor of HDACs with antidepressant-like properties” reports the analysis of toxicity, inhibitor activity and antidepressant properties of histone deacetylase inhibitors.

The study is valuable. The authors should address the following issues.

Lines 39-45: the cited papers are focussed on learning and memory more than on psychiatric disorders, as mentioned in the text; either the references or the text should be modified to reach a better agreement.

The overall number of animals should be reported in the methods, as well as the number of the specific authorisation released by an Ethical Committee on Animal Welfare for performing this study.

The source of A. salina should be described.

Whether mice were randomised to experimental group and if experimenters evaluating behavioural scores were blind to experimental treatments should be reported.

Section 2.6.2: FST in the originally published procedure tested rats, non mice, thus some modifications were adopted by the authors. Also, why were five administrations selected? And how was the dose chosen?

Lighting conditions in the ETM should be described.

Since parametric tests were used in the statistical analysis, how normal distribution was checked should be reported.

It is not clear which software was used for curve fitting in the analysis of dose response curves, as well as how the statistical analysis was performed.

Line 221: F value seems to be a mistake.

The number of replicates in experiments reported in section 3.2 should be better explained as it is not clear if only technical replicates were performed or the experiment was repeated.

Minor:

The sentence in lines 45-49 is long and difficult to follow: the authors should consider revising it.

Line 220 : “shown” should be showed.

Author Response

Reviewer 2

The manuscript “Biochemical and behavioral characterization of IN14, a new inhibitor of HDACs with antidepressant-like properties” reports the analysis of toxicity, inhibitor activity and antidepressant properties of histone deacetylase inhibitors.

The study is valuable. The authors should address the following issues.

We really appreciate the observations and suggestions made; we believe were very useful to improve the quality of our work.

Lines 39-45: the cited papers are focussed on learning and memory more than on psychiatric disorders, as mentioned in the text; either the references or the text should be modified to reach a better agreement.

As the reviewer suggested we modified the references according to the text, lines 48-9.

The overall number of animals should be reported in the methods, as well as the number of the specific authorisation released by an Ethical Committee on Animal Welfare for performing this study.

As suggested by the reviewer, we added the total number of animals used. Please refer to the first sentence of the Materials and methods section. The number of authorization was already mentioned in the same section, lines 83-5: “Experiments were performed in accordance with the protocols approved by the Committee on the Use of Live Animals in Teaching and Research of the UNAM (FM/DI/036/2017), which comply with the "International Guiding Principles for Biomedical Research Involving Animals".”

The source of A. salina should be described.

As suggested by the reviewer the source of A. salina is now included in line 110: “A. salina cysts were (Eclosion azul®) were obtained at a local aquarium and hatched in seawater (3%).

Whether mice were randomised to experimental group and if experimenters evaluating behavioural scores were blind to experimental treatments should be reported.

Mice were simple randomized to the treatment groups without considering any other variable. In line 83-5 we added the sentence: “The animals were simple randomized to the treatment groups, using the random number generator (Rand function) of MATLAB software, and… “. Experiments were performed in a double-blind manner, since the allocation, until the data analysis. Therefore, in lines 91-2 we added: Moreover, all experiments were performed in a double-blind manner.”

Section 2.6.2: FST in the originally published procedure tested rats, non mice, thus some modifications were adopted by the authors. Also, why were five administrations selected? And how was the dose chosen?

The observation of the reviewer is correct. We made some modifications to the original method according to Farzin and Mansouri, 2006. These modifications are included in the new version of the manuscript, along with the abovementioned reference (lines 195-6).

Why were five administrations selected?

With the aim of achieving an optimal dose of the HDAC inhibitors in the CNS, we decided to use this protocol, since it has been successfully used to administer this type of drugs in previous studies (Beconi et al 2012. PLoS One. 7(9):e44498. doi: 10.1371/journal.pone.0044498). In figure 4, it could be observed that a five-days treatment is suitable to reach a higher concentration in all fluids evaluated.

And how was the dose chosen?

We selected the dose according to a preliminary behavioral study, where we tested the same five-days treatment reported here using IN14 100, 200 and 400 mg/kg/day; the results of these experiments showed that all three doses had similar effects on spontaneous locomotor activity and the FST. Therefore we decided to use the lower effective dose in this work. We have added a sentence in the manuscript to clarify this point. Please refer to the 2.6.1 “Treatment” section.

Lighting conditions in the ETM should be described.

As suggested by the reviewer, we clarified it by adding the sentence: “Behavioral testing was carried out in a soundproof room with an illumination level maintained at 100 lux”, lines 212-13.

Since parametric tests were used in the statistical analysis, how normal distribution was checked should be reported.

We had neglected to test for normality in the previous version. Therefore, we performed D’Agostino-Pearson Normality tests in all groups. In the FST, some of the groups did not pass normality and we therefore reanalyzed the data for this experiment with Kruskal-Wallis tests followed by Dunn’s multiple comparisons tests to compare each group with the vehicle. The first paragraph of Section 3.4 was therefore re-written to include the new analysis; the overall interpretation of the experiment remains the same however, so we left the discussion section intact. Finally, groups from all other experiments did pass normality tests.

It is not clear which software was used for curve fitting in the analysis of dose response curves, as well as how the statistical analysis was performed.

As mentioned in section 2.7. Data Analysis, statistical analysis was performed using GraphPad Prism® software version 6.01. The specific analyses made are now mentioned in the manuscript: “Non-linear regression curve fit toll was used to obtain the LC50 of all compounds tested”.

Line 221: F value seems to be a mistake.

Indeed, the F value reported in the previous version was a mistake. The correct value is F (5, 12) = 880.7, p<0.0001. The made the correction in the new version (line 255).

The number of replicates in experiments reported in section 3.2 should be better explained as it is not clear if only technical replicates were performed or the experiment was repeated.

Certainly, the triplicates mentioned in the manuscript were technical replicates, since we performed an in vitro inhibition assay, in which the manufacturer provides the chemicals, including: substrate, enzyme (HeLa nuclear sample), developer and TSA inhibitor as control. Therefore, the replicates were used to minimize variability. We added the new inhibitors and PB as positive control. This was clarifying in the text; see section 2.4. “In vitro ELISA-Based HDAC Activity Assay”.

Minor:

The sentence in lines 45-49 is long and difficult to follow: the authors should consider revising it.

As the reviewer suggested, we replaced the sentence: “One of the major categories of epigenetic biochemical mechanism is histone post-translational modification of which histone acetylation - the addition of an acetyl group in the N-terminal lysine residues in the nucleosomal core of histone proteins [8,9] - has been highly implicated in neuroplasticity [10].”

…from the earlier version, with the shorter:

“Amongst epigenetic mechanisms, histone acetylation - the addition of an acetyl group in the N-terminal lysine residues in the nucleosomal core of histone proteins [8, 9] - has been highly implicated in neuroplasticity [10].”

Line 220 : “shown” should be showed.

We replaced the word “shown” with the word “showed” in the new version (line 254).

Reviewer 3 Report

The manuscript describes the evaluation of 3 compounds, previously designed and synthesized by the authors, as class I HDAC inhibitors. One of them, IN14 possessed a better profile than the other two and its pro-cognitive and antidepressant effects were evaluated on animal models.

The use of animals was approved from the ethics committee of the University and is stated that efforts were taken to minimize pain and discomfort to the animals while conducting these experiments.

The manuscript can be accepted after minor revision based on the following comments:

A figure showing the structures of the tested compounds should be added

Purity data of the tested compounds were not clearly reported in the previous manuscript of the authors (ref 19) for compound IN14 while for the 2 others these data are missing.

The known antidepressant DMI, used as control, was tested at lower dose. The authors should explain this.

Author Response

Reviewer 3

The manuscript describes the evaluation of 3 compounds, previously designed and synthesized by the authors, as class I HDAC inhibitors. One of them, IN14 possessed a better profile than the other two and its pro-cognitive and antidepressant effects were evaluated on animal models.

The use of animals was approved from the ethics committee of the University and is stated that efforts were taken to minimize pain and discomfort to the animals while conducting these experiments.

We really appreciate the observations and suggestions made; we believe were very useful to improve the quality of our work.

The manuscript can be accepted after minor revision based on the following comments:

A figure showing the structures of the tested compounds should be added.

As suggested by the reviewer the figure was added to the manuscript in section: 2.2. Chemicals and reagents.

Purity data of the tested compounds were not clearly reported in the previous manuscript of the authors (ref 19) for compound IN14 while for the 2 others these data are missing.

Certainly, the reviewer is right. Neither in the article published last year nor in this manuscript purity of the compounds was reported. However, using thin layer chromatography and spectroscopic techniques (1H and 13C quantitative Nuclear Magnetic Resonance (qNMR) and Electrospray Ionization (ESI) high resolution mass spectrometry), the compounds were characterized and the yield and purity were determined. IN01: 95% yield, 95% purity; IN04: 35% yield, 80% purity; and IN14: 95% yield, 98% purity. We include this information in the new version of the manuscript. Please refer to section 2.2 “Chemicals and Reagents”.

The known antidepressant DMI, used as control, was tested at lower dose. The authors should explain this.

The dose of desipramine is rather in the higher range compared to earlier reports and was based on a previous study using the CD1 strain in the FST task (Costa et al, Prog Neuro-Psychophram Biol Psych, 2013, http:// dx.doi.org/ 10.1016/ j.pnpbp.201 3.05 .002).

Reviewer 4 Report

  The present manuscript shows the capacity to inhibit HDAC activity and antidepressant-like activity of three published class I HDAC inhibitors namely IN01, IN04, and IN14, using the toxicity test of Artemia salina, in vitro fluorometric HDAC activity assay and forced-swimming test, respectively. Furthermore, the pro-cognitive and antidepressant effects of IN14 were evaluated since it was more potent than the other two compounds. the data is promising, since the test of forced swimming model of depression, intraperitoneal administration of IN14 (100 mg/Kg/day) during 5 days decreased immobility, a  marker of behavioral despair significantly more than tricyclic antidepressant desipramine, while also increasing climbing behavior, a putative marker of motivational behavior. moreover, IN14 left the retention latency in the elevated T-maze unaltered. this new compound HDAC class I inhibitor IN14 can be a promising new antidepressant with low toxicity. more studies on this compound are required.

comments: please revise the English language carefully

Author Response

Reviewer 4

The present manuscript shows the capacity to inhibit HDAC activity and antidepressant-like activity of three published class I HDAC inhibitors namely IN01, IN04, and IN14, using the toxicity test of Artemia salina, in vitro fluorometric HDAC activity assay and forced-swimming test, respectively. Furthermore, the pro-cognitive and antidepressant effects of IN14 were evaluated since it was more potent than the other two compounds. the data is promising, since the test of forced swimming model of depression, intraperitoneal administration of IN14 (100 mg/Kg/day) during 5 days decreased immobility, a marker of behavioral despair significantly more than tricyclic antidepressant desipramine, while also increasing climbing behavior, a putative marker of motivational behavior. moreover, IN14 left the retention latency in the elevated T-maze unaltered. this new compound HDAC class I inhibitor IN14 can be a promising new antidepressant with low toxicity. more studies on this compound are required.

We really appreciate the observation; we believe was very useful to improve the quality of our work.

comments: please revise the English language carefully

The English language was checked carefully again.

Round 2

Reviewer 1 Report

The authors have addressed all my concerns. 

Reviewer 2 Report

The authors have addressed my concerns